

# The Earth's climate system recurrent & multi-scale lagged responses: empirical law, evidence, consequent solar explanation of recent CO₂ increases & preliminary analysis

Jorge Sánchez-Sesma[1,2]

[1]Instituto Mexicano de Tecnología del Agua, Jiutepc, Morelos, 62550, Mexico
[2]Now, independent consultant, Cuernavaca, Morelos, 62440, Mexico

*Correspondence to*: Jorge Sánchez-Sesma (jorgesanchezsesma@yahoo.com)

**Abstract.** This paper analyzes the lagged responses of the Earth's climate system, as part of cosmic-solar-terrestrial processes. Firstly, we analyze and model the lagged responses of the Earth's climate system, previously detected for

geological and orbital scale processes, with simple non-linear functions, and we estimate a correspondent lag of ~1600-yr for the recently detected ~9500-yr scale solar recurrent patterns. Secondly, a recurrent and lagged linear influence of solar variation on volcanic activity and carbon dioxide ($CO_2$) has been assessed for the last millennia, and extrapolated for future centuries and millennia. As a consequence we found that, on one side, the recent $CO_2$ increase can be considered as a lagged response to solar activity, and, on the other side, the continental tropical climate signal during late Holocene can be

considered as a sum of three lagged responses to solar activity, through direct, and indirect (volcanic and $CO_2$), influences with different lags of around 40, 800 and 1600 years. Thirdly, we find more examples of this ~1600-yr lag, associated with oceanic processes throughout the Holocene, manifested in the mineral content of SE Pacific waters, and in a carbon cycle index, $CO_3$, in the Southern Atlantic. Fourthly, we propose the global ocean circulation processes, that include the well known meridional overturning circulation, and the thermohaline circulation, as a global mechanism capable of explaining the

lagged forcing (volcanic activity & $CO_2$) and continental tropical climate responses to solar activity variations. Finally, some conclusions are provided for the lagged responses of the Earth's climate system with their influences and consequences on present and future climate, and implications for climate modelling are preliminarily analyzed.

## 1 Introduction

*"In a short preliminary review of the scientific work of the Norwegian-British-Swedish expedition, E.F. Roots points out that*

*the lag between climatic change and change in form of the glaciers may be longer than the period of climatic change itself."*

*Glacier Variations and Climatic Fluctuations (Ahlman, 1953)*

There are several indications that the Solar System is in fact an open system, strongly interacting with its cosmic environment (Rampino and Stothers, 1985; Rampino, 1997; Shaviv, 2002). As a consequence of these indications, the Earth's climate system



has also been considered an open system influenced not only by its internal forcing processes (volcanic activity, carbon dioxide and human activities), but also by galactic activity and solar activity (Pexioto and Oort, 1992). Additionally, all these influences are transmitted and linked through different and complex geophysical processes, with delayed responses in different temporal scales (van Andel, 1994; Shackleton, 2000; van de Plassche et al., 2003).

In order to analyze the Earth's climate system lagged response to different forcings, in a previous work we have analyzed and forecast solar activity over millennia scales (Sánchez-Sesma, 2016; SS16 hereafter). Considering that solar activity, volcanic activity and carbon dioxide are the most important forcing processes and variables of the Earth's climate system, since all of them, or their interactions and/or byproducts, generate, modulate and modify radiation processes in the atmosphere, in this work we develop a joint study of their origins, looking for insight on long-term behavior and possible common mechanisms of their

processes, feedbacks and recurrent influences.

In the following section we will introduce an empirical analysis and modeling of the lagged influences of solar activity on the main forcing and processes of the Earth's climate system (volcanic activity, $CO_2$ and tropical climatic indexes), and we will provide the justification, motivation and structure of this work.

**1.1 The forcing of the Earth's climate system**

Here we introduce solar activity, volcanic activity and $CO_2$, and we emphasize their characteristics and long-term delayed influences and responses (mainly in multi-centennial & millennial-scales).

a.   *Solar Activity.* Its values are around 1366 watts per square meter in the top of the atmosphere, providing Earth's climate system with energy through IR, UV and many other electromagnetic wave lengths. Its long-term recurrences, persistence and variability, and its possible astronomical forcing processes, have been analyzed in SS16.

b.   *Volcanic Activity.* It influences Earth's climate system because generates sulfate aerosols that reduce incoming solar radiation, warming the stratosphere and altering ozone creation, reducing global mean surface temperature, and suppressing the hydrological cycle. Although large volcanic eruptions inject enough material into the stratosphere to significantly affect the climate system in transient events with large inter-annual to decadal-scale changes, their role in longer-term (multi-decadal to centennial) modification of the Earth's climate system is still in debate (Robock, 2015). On the other hand,

geophysical and astronomical forcings have been considered as the cause of volcanic activity (Robock, 2000). For instance, volcanic activity has been potentially considered as a response to extreme cooling periods, or to a common geophysical or astronomical cause of both processes (Rampino et al., 1979). A related study of this cooling-volcanism correlation focused on enhanced hydrothermal activity during the last two glacial terminations (Lund et al., 2016), is suggesting that: "…glacial maxima and lowering of sea level caused anomalous melting in the upper mantle and that the subsequent magmatic

anomalies promoted deglaciation through the release of mantle heat and carbon at mid-ocean ridges."





c.  *Carbon Dioxide.* The carbon cycle gases (mainly $CO_2$, O3 and CH4), constitute, jointly with $H_2O$, the most important GHGs. As the greatest "player" of these gases, $CO_2$ is considered the most important substance in the biosphere, serving as the source of carbon to the existence and development of life (Revelle, 1985). On the other hand, the carbon cycle, expressed with carbonate ion $[CO_3^{2-}]$ concentration releases from the deep ocean at glacial terminations and deglacial processes, is a critical component of past climate change, although the underlying mechanisms remain poorly understood (Yu et al., 2014).

d.  *Human Activities.* In addition to previously commented influences on climate, recent studies have highlighted the importance of human activities with increased anthropogenic influences [$CO_2$, CH4, deforestation, change of soil use, etc.]) on global warming (GW) are currently active, imposing a projected change of $4 \pm 2$ °C by 2100 AD that seems to exceed the maxima values estimated for the past millennia (IPCC, 2013).

## 1.2 The resonant environment of the Earth's climate system

Based on recently reconstructed paleoclimate records for solar-related variables, SS16 has found a ~9500 yr recurrence of SA. SS16 has also analyzed the recurrent planetary gravitational forcing of the solar activity that contributes to a terrestrial resonant environment. However, there are also important works by many researchers, providing a theoretical basis and practical evidences of resonant interactions of cosmic with terrestrial processes, for instance:

a.  Focused on Moon-Earth gravitational links, Keeling and Whorf (2000), based only on the links expressed in tidal astronomical periodicities, have proposed a ~1.8 Kyr that represents the time for the recurrence of perigean eclipses closely matched with the time of perihelion. This cycle is near the fifth part of the ~9.5 Kyr detected periodic recurrence. Keeling and Whorf (2000) also detected in their analysis a ~4.65 Kyr modulation cycle of the 1.8 Kyr that is almost half of the ~9.5 Kyr detected solar recurrence. For instance the ~4.65 Kyr lunar forcing could be associated through a (3:2) resonance with the recently detected ~7.0 Kyr climate and sea-level oscillation during Marine Isotope Stage 3 (Clark et al., 2007).

b.  We also compared the equatorial insolation variability recently evaluated by Berger et al. (2006), who, in line with Milankovitch ideas on astronomical forcing, and using astronomical models and multi-tapper method of spectral techniques, evaluate significant 95 Kyr and 123 Kyr periods related to eccentricity periods. The lowest value of 95,000 yrs is almost ten times the time-scale of the solar recurrence detected. This also means that one of Earth's primary eccentricity periods is in resonance (10:1) with the solar periodicity detected.

Other examples of these Earth's climate system resonant responses to solar recurrent forcing are found in the paleoclimatic literature. For instance, Pestiaux, Duplessy and Berger (1987), after an analysis of four deep-sea cores from the Indian Ocean that is influenced by the monsoon circulation, estimate significant spectral maxima located at 10.2 +/- 1.2, 4.6 +/- 0.3 and 2.3+/- 0.2 Kyr, roughly including harmonics of themselves and of the 41, 23 and 19 Kyr periodicities found in insolation (Berger and Pestiaux, 1984). However, when we select only the two cores of the Indian Ocean NE corner, more isolated from the Southern



ocean flows than the other two cores, the maxima were located at 9.5 +/- 2.1, 4.86 +/- 0.77 and 2.53+/- 0.16 Kyr, including the detected ~9.5 Kyr solar recurrence and its harmonics.

Another important example of these possible Earth's climate system resonant responses to solar recurrent forcing is provided by Rial and Yang (2007), who, looking for abrupt climate change evidences in Greenland ice cores from the last 100 Kyr, and after

applying a high pass filter to GRIP time series, detected significant spectral peaks located at 9.3, 6.2, 4.6, 3.6 and 2.97 Kyr, close to the detected ~9.5 Kyr solar recurrence and its harmonics.

### 1.3 The Earth's climate system lagged responses

Due to changing forcing (solar activity, volcanic activity and $CO_2$) and ocean dynamical processes, the Earth's climate system has been changing in different temporal scales.

As the climate does, sea level (SL) changes occur over a broad range of temporal scales, and SL is strongly correlated not only with climate but also with the diversity of life (van Andel, 1994). For instance, SL has changed over geologic time up to hundreds of meters. As Hallam's (1984) Phanerozoic SL reconstruction has shown, SL has changed with a maximum relative SL of +400 meters above the present SL in oscillations of hundreds of million years.

Although SL change has been considered to be a result of many contributing processes, it can be considered as an integral

measure of climate change (Milne et al., 2009), with all its complexity included.

Is important to consider that the primary contributors to contemporary SL change are the expansion of the ocean as it warms and the transfer of water currently stored on land to the ocean, particularly from land ice (glaciers and ice sheets) (Church et al., 2013). However, numerical experiments with an ocean-atmosphere coupled model have shown that centennial variations in the SL in the NW Atlantic may be associated with lagged influences of solar-forced variations of deep-ocean salinities and

temperature in the North Atlantic processes of the global ocean circulation (GOC) (van de Plassche et al., 2003; hereafter vdP03).

All of these complex aspects of sea level & climate variations and lagged responses may be better understood if we analyze this phenomenon with a multi-scale multi-process approach, based on long-term scale information (Lovejoy and Schertzer, 2013), and including multi-scale lagged responses.

### 1.4 The motivation and description of this work

The motivation of this work is to contribute to a better understanding of climate change. The knowledge of these paradigmatic issues remains incomplete for a number of reasons.

Firstly, paleoclimate studies have just begun to consider cosmoclimatic approaches that include galactic cosmic rays and cosmic dusts (Shaviv et al., 2014). Secondly, the solar activity, volcanic activity and $CO_2$ forcing of climate at centennial and millennia

scales have not been jointly analyzed, neither forecasted, and require further work, with novel methods and updated data. For





instance, although low and high solar activity has been linked to cooling during the Little Ice Age (AD 1450–1850) and regional warmth during the Medieval Climate Anomaly, respectively, it was found that the amplitude of the associated solar changes is poorly constrained with estimates of solar forcing spanning almost an order of magnitude (Schurer et al., 2014). Thirdly, the detected Earth's climate system lagged responses (found in previous studies and integrated, analyzed and empirically modelled

in this work) in different scales, processes and regions of the world, modifies our conceptions of and approaches to modelling sea level & climate change processes. And fourthly, solar influence on the Earth's climate system appears to be advected and modulated (as will be shown in this work) by the Atlantic component of GOC (AGOC, or the well known AMOC) that increased during the present interglacial (Piotrowski et al., 2004).

Following SS16, where simple recurrent models of solar activity signals were applied and tested, suggesting the existence of

multi-millennial (~9500-yr) scale solar oscillations, and Haigh (2011), who pointed out that "it is now possible to identify decadal and centennial signals of solar variability in climate data," we have initiated in this work a systematic analysis, and consequent forecasts, of the recurrent and lagged influences of solar activity in the Earth's climate system. Firstly, we model the non-linear lagged responses of the Earth's climate system with a simple power law (Appendix A). Secondly, we extrapolate modulated, lagged and recurrent influences on volcanic activity, $CO_2$ and continental tropical climate (CTC). Thirdly, we verify

the power law for the response to the 9500 yr solar pattern of forcing with three climate cases (Appendixes A): a) Iron deposition in the south-western Pacific lags solar activity by ~1600 yrs, and b) atmospheric $CO_2$ variability lags solar activity by ~1600 yrs. And finally, we discuss the lags detected, the consequent forecasts, and their implications in the climatic models.

## 2 Methodology

### 2.1 Data

In order to analyze the lagged responses of the Earth's climate system to recurrent solar oscillatory patterns, we use information coming from several reconstructed solar and climate proxy records.

Firstly, we employ the three solar reconstructed records of total solar irradiance (TSI) established by Steinhilber et al. (2009; hereafter S09), Steinhilber et al. (2012; hereafter S12), Solanki et al. (2004; hereafter S04). These records, presented and analyzed in SS16, are depicted in Fig. 1a.

Secondly, we select volcanic activity and $CO_2$ reconstructed records for the last millennia, coming from different proxy records. The volcanic activity index record is based on Antarctic EPICA dome C ice core sulphate densities (Castellano et al., 2005). The $CO_2$ proxy record is the carbonate ion content in SW Atlantic deep sediments, near the border of the Antarctic circumpolar current for the last 28 Kyr (Yu et al., 2014). Complementarily, the EPICA atmospheric $CO_2$ record was selected to be jointly analyzed with solar activity records for the deglacial and Holocene periods. These three records are depicted in Fig. 1b and 1c,

and in Appendix A, respectively.



Thirdly, we look at a climate record for a tropical area for the last millennia, specifically the Congo River basin surface air temperature (CRB-SAT) record, because it covers the last 25 Kyrs and it is a continental tropical climate record obtained with a novel and promising technique (W07), based on molecular changes in lipids associated with surface temperatures. This CRB-SAT record (W07), or Tcrb, is relatively more solar-influenced than in non-tropical areas of the world, because the latitudinal

distribution of solar influences shows its maximum in the tropics. Moreover, this record is also relatively less influenced by the ocean than in coastal regions, because its signal is coming from the central tropical zone of Africa, specifically the CRB (the world's second largest river basin comprising an area of more than 3.4 million square km), which is also relatively isolated by topography. The detrended Tcrb information is displayed in Fig. 1d, and presents higher oscillations in the glacial and deglacial periods with respect to those corresponding to the Holocene. Although oceanic influences on the Tcrb are minimal, Tcrb

responses to solar activity should be modulated by the GOC during the interglacial differently due to its increasing intensity in the Atlantic GOC (AGOC/AMOC) that has been reconstructed by Piotrowski et al. (2004), who pointed out: "From a minimum during the Last Glacial Maximum, North Atlantic Deep Water  began to strengthen between 18 and 17 Kyr cal. BP, approximately 2000–3000 years before the Bølling warming."

Fourthly, we select four different cases of the Earth's climate system lagged responses at geological and orbital scales. In those

responses, SL has been delayed, affected by plate collisions and orbital astronomical forcing (van Andel, 1994; Shackleton, 2000). While the geological lags are due to mass inertia and volumetric readjustments of continents (van Andel, 1994), the orbital scale lags are due mainly to the thermal and mass inertias associated with ice melting of glaciers and polar ice-sheets (Shackleton, 2000). These four pairs of period/lag (P/L) samples of the climate oscillation/lagged response processes are shown in Fig. 2 and Appendix A (Table A1).

And fifthly, we select a Holocene climate record from important oceanic variable related to the chemistry of the ocean. It was iron deposition in the SW Pacific, off the Chilean coast, a reconstructed record by Lamy et al. (2001), which is presented and analyzed in Appendix A.

## 2.2 Modelling

We employ three models to analyze the Solar-terrestrial recurrent & lagged connections.

The first model supposes an inverse connection between Tcrb variance and AMOC intensity. It is based on modeling results in the North Atlantic by Shakun et al. (2012), who estimated an increase of around 100% of AMOC intensity (from around 4 to 8 Sv) during the deglacial period (from 17 to 10 Kyr BP), corroborated with proxy-based AMOC reconstructions. This first model for the lagged and modulated linear contribution of a proxy variable is proposed as follows:

$$SC(t) = M[\alpha_P P(t+\delta_P) + \beta_P(t-t_1) + \gamma_P] + e_P(t), \qquad (1)$$

With M=1 for 0<t<10KyrBP, M=1+0.133(t-10) for 10<t<17.5KyrBP, and M=2 for 17.5<t<25KyrBP.





Here, $P(t)$ is the proxy variable, $\alpha_P$ is the amplification factor, $\beta_P$ is the slope, $\delta_P$ is the lag, $\gamma_P$ is the additive constant, $t_1$ is the initial times for the modeled period, and $e_P(t)$ is the error of this model.

The second model is obtained through modifying eq. 1 without considering a variable modulation (M=1). It is the following:

$$SC(t) = \alpha_P P(t+\delta_P) + \beta_P(t-t_1) + \gamma_P + e_P(t), \tag{2}$$

The third model, an analogue model, is defined as:

$$SC(t) = \alpha_A SC(t+\delta_A) + \beta_A(t-t_1) + \gamma_A + e_A(t), \tag{3}$$

Here, $\alpha$ are the amplification factors, $\beta$ are the slopes, $\delta$ are the lags, $\gamma$ are the additive constants, $t_1$ are the initial times for the modeled period, and $e(t)$ are the analogue error of these models. The subindexes P and A indicate proxy and analogue based models, respectively.

In all these models, parameters are estimated through iterative or multi-linear regression processes that minimize the RMS values of errors. Taking into account both the dating limitations and the approximated values provided by proxy reconstructions, and, instead of developing statistical analysis as convergence and confidence level estimations, we prefer in this stage of research on climate recurrences, to apply verification/replication of all of our findings with independent information in our estimation processes and results. Future climate reconstructions with more accurate information will provide further and refined

statistical analysis.

## 3 Results

### 3.1 A model of the lagged response of the Earth's climate system

In Appendix A we propose and apply a model of the Earth's climate system lags for different forcing periods. Three power law models (Avg.-Std.Dev., Avg., and Avg+Std.Dev) were adjusted to evaluated information, previously commented, coming from

Shackleton (2000) and van Andel (1994). These models are displayed in Fig. 2. Based on these obtained power law equations, we have also verified two cases of SL lag analyses developed by Howard et al. (2015) and van de Plassche et al. (2003), associated with forcing periods of 12 and 650 yrs and corresponding estimated lags of ~2 and ~120 yrs, respectively. It must be mentioned that the extrapolation of the [10]Be record, required to justify the existence of a forcing period of around 650 yrs (shown in Appendix A, in Fig. A2), has also provided an additional verification of the solar forecast for future centuries

proposed by SS16.





### 3.2 The lagged response of the Earth's climate system to the ~9500 yr solar activity recurrent pattern

Based on the obtained power law equations (see Fig. 2), we are able to estimate the lag corresponding to the TSI ~9500 yr recurrent patterns. Following these power laws, this lag must be located within a probability of 63%, in the range from 1270 to 2120 yrs and with a central value of 1695 yrs. It is shown in Fig. 2 with vertical and horizontal red lines.

In Appendix A we analyze different examples of this type (and size) of sea level & climate lagged responses. On the one hand, in Appendix A we analyze iron deposition in the south-western Pacific ocean waters and the Antarctic EPICA $CO_2$ variations as a lagged response of ~1570 and ~1670 yrs to solar activity, respectively.

Also, in Appendix A, we have analyzed, with spectral analysis tools, the AGOC/AMOC "natural variability" data simulated in a control run, forced only by the annual cycle, provided by the climate-ocean group of the Max Planck Institute, and one range of
main periodicities (significant level of 1%) is defined from 2600 to 3200 yrs. We consider that these oscillation periods of the natural North Atlantic GOC (AGOC/AMOC) modes that are near the double of length of the 1600 yr lags could generate a kind of resonances between lagged responses and oceanic recurrent oscillations.

### 3.3 Volcanic activity lags solar activity by 780 yrs

The first solar-terrestrial connection to be analyzed is the solar-volcanic. It is detected by analyzing solar and volcanic records.
The results depicted in Fig. 3 show the solar negatively-correlated lagged influences on volcanic activity, with a lag of around 780 yrs.

A verification of the influence of the ~9500 yr solar activity recurrent patterns in the Earth's climate system is provided in Fig. 3b, which shows a 9480 yr recurrence trend of volcanic activity similar to those presented by solar activity.

### 3.4 A marine index of carbon cycle lags solar activity by 1550 yrs

Another important solar-terrestrial connection to be analyzed here is the solar- $CO_2$, and it is detected based on solar and carbon proxy records. The results, displayed in Fig. 4a, show that the linearly detrended marine index, $CO_3^{2-}$, lags solar influences by 1550 yrs. The lag is close to the other Antarctic EPICA $CO_2$ variations that lag solar activity by ~1670 yrs (Appendix A).

Fig. 4b shows two recurrent trends of the $CO_3^{2-}$ index, with 9500 and 19000yr lags, similar to those presented by solar activity, also displayed in the same Figure.

### 3.5 Cross correlation: Solar activity versus tropical climate (TSI.vs.Tcrb)

Another solar-terrestrial connection to be analyzed is the solar-tropical climate. It is evaluated with the cross correlation between the TSI (S04) lagged record and the continental tropical climate record Tcrb (W07), during the late Holocene. The results displayed in Fig. 5 show the maxima cross-correlations at solar lags of 40, 1650 and 3300 years. The same figure shows a model





integrated with the simple sum of two periodic sine oscillations with periods of 820 and 1600 years, explaining the 76.3% of the analyzed cross correlation record over the last three millennia.

A verification of the multiple lagged solar influences, directly on forcing (volcanic activity and $CO_2$), and indirectly on tropical climate (Tcrb), are obtained when the Tcrb is linearly decomposed in two functions based on solar activity. These functions, one based only in a lagged TSI signal by 40 yrs, and other based in three lagged TSI signals by 40, 780 and 1600 years, provide explanations of the Tcrb variance of 32 and 62%, respectively. These functions, or simple models, and the tropical climate (Tcrb) signal are depicted in Fig. 6.

### 3.6 Influences of the ~9500 yr recurrence of solar activity on continental tropical climate

Three models of the continental tropical climate (CTC) temperatures, Tcrb, only based on different TSI recurrent reconstructions that employed Eq. 1, are displayed in Fig. 7. The modelling required a different modulation for the first (M=2.) and second (M=1.) halves to distinctly consider the increasing GOC induced deglaciation process until the stabilized Holocene periods (Piotrowski et al., 2004; Shakun et al., 2012). These models of Tcrb, which were based only on solar, S04, S09 and S12, records, explain 30.0, 23.6 and 31.6 %, and 6.5, 10.9, and 8.5 % of the reconstructed Tcrb record for the periods from 20 to 10, and from 10 to 0 Kyr BP, respectively. These three modelling results constitute other tests of our recurrent model of solar activity (SS16) and its direct influences on the ECS.

Note that the variance explanation is larger in the first half of the record when the GOC was low. This will be further discussed later.

### 3.7 A millennia scale experimental forecast of continental tropical climate (CTC) based on recurrent patterns

For another confirmation of recurrent solar activity and its influences on the Earth's climate system, we also apply equation 3 with a lag parameter of 9600 yrs to the continental tropical climate (CTC) Tcrb record. Our analog model explains most of the variation of the Tcrb during the past centuries. In a comparison, partially depicted in Fig. 8, the Tcrb analogue model explains 7.3, 18.7, 60.8 and 71.7 % of the Tcrb reconstructed record (W07) for the last 10, 5, 2 and 1 Kyr, respectively. For the future, our model provides an estimation of a cooling for the 21$^{st}$ century of about 0.5°C, followed by a slow warming trend with small oscillations during more than four centuries. The forecasts comparison also considers two different forecasts of TSI, shown in SS16 (Fig. 4). With this comparison, we estimated that the Tcrb analogue model also explains in the 2050-2500 AD period 34.3 and 37.1 % of variance of the TSI forecasts, based on S04 and S12 records, respectively.



## 4 Discussions

We are going to discuss: a) the cosmic-terrestrial resonant environment, b) the Earth's climate
system lagged response model, c) the Earth's climate system lagged responses to the ~9500 yr recurrence of solar activity both
in climate forcing (volcanic activity and $CO_2$), and also in climate variables and processes, d) its possible mechanisms and
connections with the GOC processes, and e) its implications in climatic modeling and research.

### 4.1 A cosmic-terrestrial recurrent and resonant environment

We completely agree with the following excerpt from Peterson (1993)'s excellent book "Newton's clock":

> "*Sweeping around the sun along a grand loop….nine planets respond not only to the enduring attraction of the sun but also to that of their neighbors…..causing a jangle of deviations from perfect geometry. In this intricate, discordant symphony of the planets, the giants Jupiter and Saturn call out most loudly. Mercury, Venus, Mars, Uranus, Neptune and Pluto contribute quieter voices….the lesser objects of the solar system—asteroids, satellites and comets—add to the celestial chorus, as does the thrum of the fluttering solar wind of accelerated particles and radiation continually erupting from the sun. At the same time, the globe on which we live and wonder spins on its own axis even as it orbits the sun. Like a gargantuan twirling top, it wobbles and tilts. It shudders with every earthquake and twists fitfully with every giant swirl within its atmosphere or seas. Any unevenness in its shape or in the distribution of materials making up its crust unbalances its movements and provides a lever by which the sun and other bodies can further wrench Earth from a pure and simple motion…This modern, remarkably detailed picture of solar system dynamics represents an astonishing triumph of human reasoning.*"

All of these commented astronomical characteristics of the Earth, Moon, the Sun, the other planets, and other smaller objects
have clearly shown, after many studies conducted during the last centuries, a recurrent solar system environment.

On the one hand, the daily and annual cycles are the greatest examples. On the other hand, the low-frequency variations of
planetary forces over the Earth and its internal rotational dynamics generate long-term changes in its orbit, the orientation of its
axis of rotation, with consequent changes in precession, inclination and eccentricity that generate regular cyclic variations with
periods of around 26, 41 and 100 Kyr, respectively.

Between these short- and long-term cycles, there are, recently-analyzed, millennia-scale astronomical processes. Specifically, in
a previous work, SS16 discovered and analyzed one of these variations, the 9.5 Kyr solar recurrent patterns. This recurrent solar
activity forcing over the Earth induces other forcing and responses of components of the Earth's climate system.

As the Earth's climate system includes almost all existent geospheres: the atmosphere, lithosphere, hydrosphere, cryosphere and
magnetosphere, it presents important interactions that appear to generate volcanic and $CO_2$ forcing with complex responses. For
instance, the sea level variation depends not only on the temperature and volume of the seas, but also depends on other



numerous factors, such as: melting of ice and ice-sheets, isostacy of the continents, the Earth's rotation speed, and sea currents (Gomez et al., 2015).

Additionally, in the millennia scale there is a resonant environment (see sect. 1.3) where solar variability could be enhanced by other short- and long-term Earth-Moon influences manifested clearly with regular motions and processes associated with lunar phases and eclipses, and with marine tides and changes in rotation speed (length of the day) (Keeling & Whorf, 2000).

Also, in Appendix A we have shown that there are natural oscillations in the Earth's climate system, expressed through the most recent GOC modeling, from years and multi-decadal, to multi-centennial and multi-millennial scales, which appear in the multi-millennia control runs of the OA-GCM (Jungclaus et al., 2010). Parts of these cyclic circulations should be related to the detected lags of 2, 120 and 1500 yrs of the SL response to solar forcing periods of 12.6, 650 and 9500 yrs, respectively. For instance, these lags are approximately half of the main periodic modes of natural oscillation in the AGOC/AMOC, with three of the period ranges of 4-7, 180-260, and 2600-3100 yrs, shown in Fig. A3, and indicating a potential resonance between lagged responses to solar forcing and harmonics of internal oscillation of the Earth's climate system.

It should be mentioned that recent studies have presented more detailed examples of regularities and resonances in the Milky Way realm and solar-terrestrial connections, as follows:

a. In May 2016, NASA space scientists discovered four planets in the Kepler-223 star system. The finding was helped by orbital resonance, where the four planets have regular, periodic gravitational pulls on each other, with orbital periods related by distinct ratios. In the Kepler-223 star system, located more than 6,000 light-years away, the two innermost planets are in a 4:3 resonance, and the second and third are in a 3:2 resonance (Mills et al, 2016).

b. Solar wind fast streams emanating from solar coronal holes cause recurrent, moderate intensity geomagnetic activity at Earth (Tsurutani et al., 2006).

## 4.2 A power law model of the lagged responses of the Earth's climate system

We have found a simple power law model for multi-scale climatic processes based on estimated lags in different climate scales (see Appendix A). The existence of different inertias (thermal, rheological and mechanical) and long-term transport times due to slow oceanic movements, are the basis of a non-linear multi-scale lagged response, expressed in a power law model.

These lagged responses of the Earth's climate system that are present in different scales and different variables of the oceanic and atmospheric realms, have recently began to be analyzed, and modeled. For instance, recent modeling efforts on the interaction of the Atlantic Ocean and the atmosphere, and the persistence or memory of Atlantic Ocean anomalies, have produced a simple mechanistic model of the North Atlantic climate response to solar UV variability with a lag of 2-4 years (Scaife et al., 2013).





Other longer Earth's climate system lagged responses are detected using models calibrated against a range of coupled atmosphere-ocean GCMs. For instance, Wigley (2005) finds substantial increase in global mean surface temperature in response to a present atmospheric composition held constant continuing on timescales of 50 to 400 yr. Other examples of the SL response obtained with observational and modeling information were commented upon in Appendix A.

### 4.3 The Earth's climate system response to the 9500 yr solar activity recurrent pattern

Based on the power law model of the Earth's climate system lagged responses, we estimated a lag period, of 1695 +/- 425 years, associated with the ~9500 yr oscillation, which is shown in Fig. 2.

We have found several variables (Antarctic EPICA $CO_2$, South Atlantic $CO_3$ and Iron deposition in SW Pacific) that present lags of around 1600 yr with respect to SA, and appear to be direct or indirectly linked with ocean dynamics. In the following paragraphs we discuss some particular aspects of these Earth's climate system lagged responses, which are displayed in Table 1.

*Carbon and geochemical cycle variables.* Three variables from oceanic and atmospheric realms have been analyzed with respect to the solar activity signal. The oceanic Fe from iron deposition in the SW Pacific, off the Chilean coast, atmospheric $CO_2$ from Antactic EPICA ice-cores, and marine $CO_3$ from the Southern Atlantic deep waters, have presented a similar lagged response of around 1600 yrs, to the solar signal. See Appendix A and Table 1.

*Volcanic activity.* volcanic activity presented a lagged negative correlated response with a lag of 780 yrs to the solar activity signal, which is close to half of the previous lags. The negative correlation is confirmed, while the frequency of volcanic events per millennium indicated that the last 2000 years constituted a period of enhanced volcanic activity (Castellano et al., 2005), while solar activity shows a decreasing trend from 2.8 to 0.8 Kyr BP (see Fig. 1). However, as volcanic activity generates atmospheric cooling, because eruption columns inject ash particles and sulfur-rich gases into the troposphere and stratosphere, decreasing the amount of sunlight reaching the surface of the earth, the lagged volcanic activity generates a correspondent lagged positive thermal response of the Earth's climate system to SA.

It is important to mention that the volcanic activity events have presented multi-scale connections with rapid climate changes. For instance, Rampino, Self and Fairbridge (1979) analyzed numerous major volcanic eruptions that coincide with cooling trends of decadal or longer duration that began significantly before the eruptions. The same authors suggest a mechanism: "variations in climate lead to stress changes on the earth's crust—for instance, by loading and unloading of ice and water masses and by axial and spin-rate changes that might augment volcanic potential."

Supporting our results on volcanic activity and climate, and plenty of other geo-connections, a recent study by Lund et al. (2016) has pointed out: "Here we present well-dated sedimentary evidence of enhanced hydrothermal activity during the last two glacial terminations. We suggest that glacial maxima and lowering of sea level caused anomalous melting in the upper mantle and that the subsequent magmatic anomalies promoted deglaciation through the release of mantle heat and carbon at the mid-ocean ridges." The analysis of all these lags and their potential mechanism is discussed later in Sect. 4.7



## 4.4 Tropical climate responses to the solar activity direct and indirect (volcanic activity & CO2) forcing

We have analyzed one important tropical climate record: the reconstructed Congo River basin surface air temperature (CRB-SAT) record, because it covers the last 25 Kyrs (Weijers et al., 2007; W07). This tropical climate variable is very important because its location, at the center of tropical Africa, generates a relative isolation to ocean influences and thus enhances solar
influences. This influence is known as continentality.

We have selected the Congo River basin surface air temperature (CRB-SAT) signal, or Tcrb, as one of the records of regional climate most influenced by the forcing of the sun. However, we have considered that this Tcrb record is a continental tropical climate (CTC) that is also influenced by volcanic activity and $CO_2$ variations.

Additionally, it must be emphasized that volcanic influences could be developed not only through blockage of atmospheric
radiation effects, but also through the deposition of ashes in CRB soils (Weijers, 2011). This local process could artificially enhance decays in CRB temperature (Tcrb).

Before analyzing the solar-terrestrial connection at the tropical belt, we look for the lags involved in this connection. To this end, we evaluate the lagged correlations between the TSI(S09) record and the CTC record over the last millennia. We employ the TSI(S09) record based only on 10Be, because the other two records, TSI(S04) and TSI(S12), consider 14C information that is
affected by another multidecadal lag, due to the carbon cycle in the atmosphere, oceans and soils (S09). See details of this aspect in SS16 (Appendix B).

The cross-correlation analysis finds the maximum Tcrb-TSI correlation when a lag of 40 yrs is applied to the TSI(S09) record (see Fig. 5). However, this Figure also shows a secondary cross-correlation peak at lags of around 1600 and 3200 yrs, possibly connected to the lagged response of forcing to the 9.5 Kyr solar recurrent pattern (see Table 1 and Appendix A).

It should be mentioned that in Fig. 5, a periodic model based only on two sine functions with periods of 0.8 and 1.6 Kyrs is depicted and explains more than 76% of the variance in the evaluated cross-correlation record. These two components correspond to the lagged solar influences expressed in volcanic activity and $CO_2$ variations.

However, a complete verification of the Solar direct and indirect delayed influences to CTC is obtained in the temporal domain of the late Holocene. To do that, direct influences are considered due to radiative processes of SA, and regional stabilization
adjustments, and indirect influences are considered due to the volcanic activity and CO2 lagged responses to solar activity. Specifically the continental tropical climate Tcrb signal model based on delayed solar direct and indirect contributions are proposed: a) Tcrb(t)= [aTSI(t-40yr)], and b) Tcrb(t)= [aTSI(t-40yr) + bTSI(t-780yr) + cTSI(t-1600yr)+ e(t)], respectively. Fig. 6 shows these two Tcrb models with, a) one and b) three components, explain 32 and 62 % of the variance, respectively.



### 4.5 The modulation of the CTC signal

With the selected CTC record, we have estimated a lagged and recurrent response of the CTC to SA. However, there is also a modulation of the CTC signal response to SA.

A possible explanation for the modulation of the CTC signal could be one considering the atmospheric response to higher

intensity in the NH radiation due to orbital influences in the Holocene and the asymmetric distribution of more continental area in the NH. The radiation increase favors the intensification of NH atmospheric processes, such as: trade winds, the Hadley cell circulation and the semi-permanent high-pressure systems. As more mass is advected from the tropics due to those atmospheric circulations, less change in tropical temperature is produced and vice versa. This modulation affects both the atmosphere and the ocean.

Another explanation of the detected modulation of the CTC Tcrb record, occurring during the interglacial, could also be linked to the increasing AGOC/AMOC that has been reconstructed by Piotrowski et al. (2004): "Neodymium isotope ratios in the authigenic ferromanganese oxide component in a southeastern Atlantic core reveal a history of the global overturning circulation intensity through the last deglaciation…It exhibits a gradually increasing baseline intensity that plateaus in the early Holocene." Regarding sudden climate changes during the interglacial, Clarke et al. (2001) have also pointed out: "Studies of deep ocean

sediments and ice cores as well as coupled climate model simulations have identified changes in the North Atlantic MOC as the probable mechanism. The warm events are thought to have occurred when the AMOC penetrated further into the Nordic sea, whereas the cold events coincided with times at which the AMOC has slackened, reducing the transport of warm water to the North Atlantic."

These two mechanisms may justify an advection process from the tropics to northern latitude regions that takes more or less heat

from the tropics when the circulations are intense or weak, respectively, and a consequent lower thermal response of the CTC. In a more reduced spatial scale, the GOC has shown a surface branch that extends almost around southern Africa that could represent, through the trade winds, the modulator of the continental tropical climate (CTC) record.

### 4.6 Verification and comparison of the forecast of the continental tropical climate (CTC)

A verification of the influence of the ~9500 yr solar activity recurrent patterns in the Earth's climate system is provided for CTC

and displayed in Fig. 8, showing around 9500-yr-recurrence trends similar to those presented by SA (SS16).

Additionally, we can compare the recurrent model for the CTC of around -0.5 °C by 2100 (Fig. 8), with the forecasted values by the models (IPCC, 2013) for the end of the 21$^{st}$ century, with values of around +2.5 and +4.5°C for the low (B1) and high (A2) $CO_2$ emission scenarios (Haensler et al., 2013). However, if we consider the total cumulative carbon dioxide for the 1990-2100 period, from all different sources for the A2 and B1 scenarios of 1862 and 983 GtC (IPCC, 2000), we can, with a linear

interpolation to the "zero" additional emission of GHG, arrive at a lower temperature anomaly of around +0.5. However, our modelling propose a -0.4 °C for the end of the 21$^{st}$ century. This comparison suggests a lower than "zero" (negative) carbon



emission scenarios of similar results that our extrapolated or recurrent forecast presented here for the CTC values of the next centuries.

## 4.7 Reanalysis of the $CO_2$ variations of the last millennia as a recurrent & lagged solar response

Although we have analyzed a $CO_2$ record over the last millennia in Appendix A, with information coming from ice-cores, we reanalyze the reconstructed records of $CO_2$, but based on stomatal densities of the last 13000 years, developed and provided by Rundgren and Beerling (1999; RB99) and Rundgren, M., Björck (2003; RB03).

This complementary analysis is conducted because the ice-core-based $CO_2$ reconstructed records present severe limitations both in dating and values (RB99, RB03) and in contrast, there is a strong link between stomatal frequency in woody plants and atmospheric $CO_2$ (Hincke et al., 2016).

In order to analyze these stomatal-based reconstructions of $CO_2$, we apply two simple models: a) a simple lagged analogue adjusted with bias and slope additive terms (Eq. 3.); and b) a linear and lagged transformation of the solar, TSI, record provided by S09 (Eq. 2). Accepting the observational times and the solar lag of 1720 years, the temporal data of the stomatal-based $CO_2$ records were adjusted using a constant bias of 40 years (possibly associated with the climatic lagged response processes in Northern latitudes, and to be discussed later in a future paper).

The resulting solar model of the $CO_2$ reconstructed record by RB99 provides 11, 13 and 90% of variance explanation (of reconstructed and observed values [NOAA, 2016]) for the last 2000, 1000 and 100 years, respectively. Our results shown in Figures 9 and 10 help us to: a) self-explain the stomatal-based $CO_2$ record with a lag of around 9400 yrs, and b) explain the last millennia of stomatal $CO_2$ as a lagged response (1720 yrs later) of solar activity expressed through solar activity (TSI [S09]) values.

## 4.8 The climate lagged response and its potential mechanisms

In order to discuss the Earth's climate system lagged response to solar activity 9.5Kyr recurrent pattern, we compare the detected lags around 1600-yr of the Earth's climate system response to solar activity and the ocean mixing time.
On one side, when we calculate the average and standard deviation of the three values estimated for this lag (See Table 1), we calculate 1597 +/- 64 for one standard deviation range. On the other side, we know from isotope studies in the ocean (Broecker, 1982) that the mixing time for ocean water is about 1600 years.

It should also be taken into account that all of the detected lags are close to one or two times the period in which the $CO_2$ signal lags the southern hemispheric temperature signal during the last glacial-interglacial cycle. For instance, Callion et al. (2003) have pointed out: "the sequence of events during Termination III suggests that the CO2 increase lagged Antarctic deglacial warming by 800 +/- 200 years and preceded the Northern Hemisphere deglaciation.… This result is in accordance with recent studies but, owing to our new method, more precise."





The importance of the GOC and specifically of the Southern ocean participation has been demonstrated in modeling and reconstruction of the carbon cycle over the ocean and atmospheric realms by several authors.

On one side, Skinner et al. (2015) have shown that deep-(>2000 m) and shallow sub-surface ocean–atmosphere 14C age offsets (i.e. 'reservoir-' or 'ventilation' ages) in the southwest Pacific decreased significantly from ~2689 and~ 1037 yrs during the late

glacial, to the corresponding Holocene values of ~1600 and 700 yrs, respectively. The same authors find that: "A comparison with other radiocarbon data from the southern high-latitudes suggests that broadly similar changes were experienced right across the Southern Ocean." These documented decreases of the 14C age the deglacial confirm several processes: a) the increase of the GOC during deglaciation, implying,  b) the decrease of deep-atmosphere 14C age from 2689 at the late glacial to 1600 years at the Holocene, and c) the connection Carbon cycle- GOC.

On the other side, Sigman et al 2010 have shown that: "..the cause of the pCO2 variation must be resolved if we are to undertand its place in the causal sucesion that produces glacial cycles…The ocean is the largest reservoir of $CO_2$ that equilibrates with the atmosphere on the thousand year timescale of glacial/interglacial changes in $pCO_2$, so the ocean must drive these changes. CO2 was more soluble in the colder ice-age ocean, which should have lowered $pCO_2$ by 30ppm, but much of this appears to have been countered by other ocean changes (in salinity and volume) and a contraction in the terrestrial biosphere. The most

promising explanations from the bulk of the $pCO_2$ decrease involve ocean biogeochemistry and its interaction with the ocean's physical circulation." This final explanation is remarking the importance of GOC processes in the global carbon cycle.

Additionally, Shakun et al.(2013) have found, for the deglacial and early Holocene periods, that "Differences between the respective temperature changes of the Northern Hemisphere and Southern Hemisphere parallel variations in the strength of the AMOC recorded in marine sediments". Following their analysis, the interhemispheric thermal "seesaw" could be explained as

an antiphased hemispheric temperature response to AMOC changes.

Considering all these previous discussions, and in order to better understand and model the existence of the lagged response of the Earth's climate system, we propose the global ocean circulation (GOC) as a primary mechanism of climate change. Specifically we hypothesize that:

*The Earth's climate system lagged responses of around 1600 yrs are associated with oceanic processes, but specifically with the*

*GOC phenomenon, because it transports, through surface waters, the climate signal that arrives in the poles from the tropics, and through deep waters, the returning climate signal from the poles to the tropics and beyond. This climate signal which is mainly generated by the solar activity forcing in the tropics is also influenced by volcanic activity and $CO_2$, which are also influenced in  multi-centennial delayed ways by solar activity.*




## 5 Conclusions

### 5.1 A proposed and verified model

We have proposed, applied and verified a simple power law model of the lagged response of the Earth's climate system to different forcing phenomena of geological, orbital and suborbital time-scales.

The Earth's climate system non-linear response appears to be frequency dependent and is verified twice, based on published data of multi-annual and centennial se level (SL) lagged responses. This non-linear, power law, model suggests, for the ~9500 yr detected solar oscillation (SS16), a millennial-scale climate lagged response (~1600 yr), which is verified with several forcings and variables of the Earth's climate system (See Table 1).

### 5.2 Detected lagged responses of the Earth's climate system (forcing and variables)

Regarding the Earth's climate system lagged responses, we have detected lagged influences on two of the most important forcings of the ECS. These forcings, volcanic activity (VA) and carbon dioxide ($CO_2$), generate lagged and positive correlated, thermal responses around 800 and 1600 yrs later, respectively, to solar activity (SA) variation.

These lagged and correlated influences on climate could generate feedbacks that potentially would be able to amplify the climatic responses to solar activity. For instance, the 1500 yr oscillation detected by Bond et al. (2001) could be explained
as forced by the interaction between the three radiative solar signals generated directly or indirectly by the Sun. The indirect influences are generated by Volcanoes and $CO_2$ that appears to be complex responses of the Earth's climate system to solar activity based on the global ocean circulation processes.

The detected lags of the carbon cycle indexes, of around 1597 ±64 yrs solar activity, and the relative lags/leads detected by Callion et al. (2003) and Shakun et al. (2012) [who report that $CO_2$ lags Southern Hemisphere temperature reconstruction by
620 ±660 yr , $CO_2$ leads global reconstruction by 460 ±340 yr, and $CO_2$ leads Northern Hemisphere reconstruction by 720 ± 330 yr] allow us to suggest that the temperatures of the southern hemisphere, global and northern hemisphere lag solar activity by around 977±724, 2057±404 and 2317±394 years, respectively.

### 5.3 A mechanisms for the Earth's climate system lagged responses

While the geological lags are due to mass inertia and volumetric readjustments of continents, and the orbital scale lags are
due mainly to the thermal and mass inertias associated with ice melting of glaciers and polar ice-sheets, we suggest that the millennia scale lags are mainly associated with the global ocean circulation processes.

We have suggested that the GOC phenomenon, together with thermal and mechanical capacities of the Pacific, Atlantic, Indian and southern oceans, could allow forcing (SA, volcanic activity, and $CO_2$) variability to affect several surface ocean



climates of the world, several centuries after its origin in the tropics. The global ocean circulation (GOC) constitute both linked "belts" of the "*Great Ocean Conveyor*" (other *GOC*) and also, sources and/or sinks of heat for the atmospheric currents that redistribute heat over the terrestrial surface of the world.

We have proposed that GOC through surface and deep oceanic currents provide the Earth's climate system with a capacity to generate lagged & moved responses due to the transport of heat and mass to both longitudinal and latitudinal directions and to different oceanic depths, covering completely all the ocean realm. These surface and deep transports of water is complemented with another GOC stage that is the mix of waters from mesoscale to micro turbulence. All these processes, together, completes the overall mix of the ocean properties to balance/distribute chemical compounds (salt, Fe, $CO_2$, etc.), mass and heat around the world ocean.

The overall oceanic mixing time, of around 1600 years, is a characteristic variable of the ocean lagged responses to solar activity recurrent patterns of around 9500 years. Its influences are obvious in $CO_2$ and Fe responses to solar activity.

In addition to GOC transport and mixing processes, there are other profound (literally) influences to the Earth's climate system. We estimate that when these mechanical and thermohaline dynamical Earth's climate system signals arrive to the high northern latitudes, the climatic influence to volcanic activity begins to work. This is supported by the previous works of

Rampino and Fairbridge on the climate-volcanism link, because the Arctic is the place where the dynamics of currents and masses, presenting sensible balances, create the most important changes in the Earth's crust, rotation, sea level, magnetism, ice-sheets and atmospheric pressure (see Gomez et al., 2015).

### 5.4 Forecast capacities and limitations

In this work, we have successfully initiated an empirical analysis of the recurrent and lagged influences of solar activity on

the Earth's climate system, associated with forcing (volcanic activity, $CO_2$), and with an important component of the ECS: the continental tropical climate (CTC).

These results have consequently enabled us to develop climate forecast capacities. By detecting and firstly evaluating recurrent and lagged Earth's climate system responses, influences and potential mechanisms, we are provided with essential elements to develop practical outputs and theoretical basis for an experimental long-range climate forecast.

Our experimental results have provided not only very useful and interesting results associated with the continental tropical climate, but also the elements to counter-balance the recent climate forecast based on normal climate simulations (coupled OA-GCM) that have presented the following dramatic scenarios: The values forecasted for the Tcrb by the models (IPCC, 2013) for the end of the 21$^{st}$ century are around 2.5 to 4.5°C for the low (B1) and high (A2) $CO_2$ emission scenarios, respectively (Haensler et al., 2013). In contrast, however, our Tcrb forecast provides not only lower forecasted anomaly

values, but a forcasted anomaly with a different sign.



Another similar situation is related with the AMOC weakening supposedly due to anthropogenic forcing. Some months ago Jackson et al. (2016) have put forward the following: "The Atlantic meridional overturning circulation (AMOC) has weakened substantially over the past decade….. We find that density anomalies that propagate southwards from the Labrador Sea are the most likely cause of these variations. We conclude that decadal variability probably played a key role in
the decline of the AMOC observed over the past decade."

## 5.5 Further forecast discussion and implications

However, the most relevant aspect of this forecast is that it presents lower values than scenarios of "lower than zero or negative emissions." This is a consequent limitation of the conventional modeling of climate, IPCC (2013), where multi-centennial scale lagged influences have not been considered.

Our results begin to show the multicentennial and millennial size of delays and the consequent importance of natural variability for future climate. Our results have presented preliminary estimations for mean lags of about one and two millennia for the SHT and NHT, respectively.

The study of these delays and verification of the proposed mechanism related to the GOC are very important activities that should be developed as soon as possible if we are to better understand climate processes and thereby improve climate
forecasts.

Thus, detection, analysis and modeling of the Earth's climate system lags are activities with paramount importance for better understanding not only of the forcings of the Earth's climate system, but also of their responses and timing.

The detected lagged responses of the Earth's climate system are important for future modeling efforts on multi-decadal and longer time scales. Our results provide specific information of important multi-centennial and millennial lagged responses
that need to be modeled and verified as a first step to develop capacities for an accurate modeling and forecasting of climate-related issues.

## 5.6 Final remarks

This paper has employed existing datasets and analyses to detect, model and assess recurrent and lagged climatic processes, providing an integrated set of empirical new evidences and rational deductions regarding the forcing, responses and mechanisms
of variability in the Earth's climate system.

In particular, based on the strong link between stomatal frequency and $CO_2$ in woody plants, we found an explanation of the $CO_2$ variability during the last millennia (including the past century), as a response to solar delayed influences (around 1700 yrs later) that is challenging the predominant current conception of the main industrial contribution to the atmospheric $CO_2$ concentrations during recent decades (IPCC, 2013).



Our findings suggest that recurrent variability may play an active role in natural climate change during the coming decades, centuries and millennia. The detected lags completely challenge our "nowcasting" climate concepts in which a change in radiation is expected to produce almost instantaneous changes in temperature.

A new approach is required in climate sciences. Instead of a geocentric (and anthropocentric) approach, what is necessary is a cosmos-nature-human centered approach in which all the external and internal forcing, and the geophysical, biological and human processes are considered in a balanced and objective way.

In order to support this new approach for climate sciences and the consequent political measures, we have tried to provide, in this work, a novel approach with robust evidences to support the required actions.

*Appendix A: Non-linear lagged responses of the Earth's climate system: A power law, its verification, and an example (Iron deposition in the south-western Pacific & $CO_2$ lag solar activity by ~1600 yrs)*

A power law model is proposed as follows:

$$L(t) = \alpha P^{\beta} , \qquad\qquad (A1)$$

Where, $L$ is the period of the lagged response of the Earth's climate system, $P$ is period of the oscillatory forcing, $\alpha$ is the amplitude factor and $\beta$ is the exponent of the power law.

Three power law models (Avg.-Std.Dev., Avg., and Avg+Std.Dev) are adjusted to Shackleton (2000)´s evaluated lags of three different orbital forcing oscillations, and another lag associated with plate tectonics (van Andel, 1994), and are displayed with the data of P/L pairs in Fig. A.1. Estimated parameters and equations are also displayed in this Fig. A.1.

Based on the obtained power law equations, we are able to estimate the lag corresponding to the TSI ~9500 yr recurrent patterns. This lag must be located, following the power laws with a probability of 63%, in the range from 1270 to 2120 yrs and with a central value of 1695 yrs.

Based also on the obtained power law equations, we have verified the cases of SL lag analyses developed by Howard et al. (2015; hereafter H15) and van de Plassche et al. (2003; hereafter vdP03). Firstly, in order to verify the forcing period of 12 yrs and the corresponding lag of ~2 yrs estimated by H15, we look for the lag that corresponds to a forcing period of 12 yrs. The lag estimated with the power laws has a mean value of 2.5 and a range from 1.7 to 3.3 yrs, which includes the value estimated by H15.

Secondly, in order to estimate the forcing period corresponding to the lag of ~120 yrs estimated by vdP03, we look for the forcing period of 640 yrs that corresponds to this lag based on the estimated power law. We analyze the 10 Be record reconstructed by Bard et al. (1997), which was employed by vdP03 in their simulated SL response to SA, based on recurrent models. We apply two simple models, a sinusoidal model with a period of 645 yrs, and an analog model with a lag of 647



yrs, which explain 17.6 and 32.7 % of the 10Be variance, respectively, and are displayed in Fig. A2, confirming the 640 yr provided by the power law model. It must be mentioned that the extrapolations of the Be10 record also provides a verification of the solar activity forecast for the next centuries developed by Sánchez-Sesma (2016).

*A verification: Iron deposition in the south-western Pacific [SWP] lags ~1600 yrs solar activity*

5  Based on the iron deposition in the SWP, off the Chilean coast, reconstructed record by Lamy et al. (2001) and the SSN record by S04, a cross-correlation analysis was developed. Fig. A3a shows a maximum correlation of 0.6 (0.71) with the ~10 yr original resolution (smoothed with 50yr-moving-averge) records where the Fe signal lags 1570 yrs the SSN record. With this lag, a linear lagged transformation was applied and a comparison of the Fe content record and its model based on SSN are displayed in Fig. A3b. The model explains more than 50% of the variance of the Fe record.

10  A similar model was adjusted to the EPICA Dome C, $CO_2$ (EDC-$CO_2$) data. The original data and its lineal and polynomial trend are shown in Fig. A4a.  The detrended $CO_2$ record (EDC-$CO_2$) and its model, a lagged linear transformation of the TSI reconstructed record, with a lag of 1670 years are shown in Fig. A4b.

In addition, we must also consider that there are natural oscillations in the AGOC, from years and multi-decadal, to multi-centennial and multi-millennial scales, which appear in the multi-millennia control runs of the OA-GCM (Jungclaus et al.,
15  2010). In order to show this, we have analyzed, with spectral analysis tools, data provided by the MPI climate-ocean group, with main periodicity ranges of 4-7, 42-55, 180-260, 350-550, and 2600-3100 yrs (significant level of 1%). See Fig. A5.

**Table A1. Lags of different variables with respect to their different astronomical and tectonic forcing periods**

| Variable | Osc.ForcingPeriod [Kyr] | Lag(Avg) [Kyr] | Lag(Avg-s) [Kyr] | Lag(Avg+s) [Kyr] |
|---|---|---|---|---|
| d18OAir | 21* | 4.6* | 4.2* | 5* |
| d18OAir | 41* | 7* | 5* | 9* |
| IceVol. | 100* | 14* | 9* | 19* |
| Sea Level | 100,000** | 30,000** | 27,000*** | 33,000*** |



**\*Shackleton (2000) and astronomical forcing,**

**\*\*van Andel (1994) and tectonic forcing,**

**\*\*\*supposing a std.dev = 0.1avg.**





**Figure A1. Multiscale empirical model of climatic lagged response. Three power law models of lags, for the average and +/- standard deviation values, are displayed. Empirical models of lags in terms of climate oscillatory periods are adjusted to four estimations made by Shackleton (2000) and van Andel (1994). See data in Table A1.**





**Figure A2. Double recurrent modelling of the Be10 record (Bard et al., 1997). A first model is based on a simple sine function with a period of 650 yrs, which explains 17% of the variance in the 10Be record. A second model is a simple analogue, which employs a linear transformation and trend adjustment with a lag of 647 yrs, explaining 33% of the variance in the 10Be record for the last four centuries. The vertical scale is inverted in order to indicate minimum and maxima of solar activity in the upper and lower parts of the graph, respectively.**





**Figure A3. Correlation and comparison between the south-western Pacific (SWP) iron deposition reconstructed by Lamy et al. (2001) and a proposed model based on the lagged influences of the solar activity SSN (S04). a) Lagged correlation values between SWP Fe deposition and TSI for different lags and two different resolutions of around 10 and 50 yrs that show a maximum at a lag of 1570 yr, and b) The model is a simple analogue, which employs a linear and trend adjustment of SSN(S04) with a lag of 1570 yrs, explaining 50% of the variance in the SWP Fe content record for the 8 last millennia.**



**Figure A4.** EPICA Dome C CO₂ (EDC-CO₂) record and a model based on the lagged influences of the solar activity TSI (S04, S09 and S12). a) The EDC-CO2 record and its trend, and b) The detrended EDC-CO₂ record and its model, which employs a linear and trend adjustment of SSN(S04, S09, S12) with a lag of 1670 yrs.



**Figure A5. (a)** AMOC ctrl. **(b)** The wavelet power spectrum. The contour levels are chosen so that 75%, 50%, 25%, and 5% of the wavelet power is above each level, respectively. The cross-hatched region is the cone of influence, where zero padding has reduced the variance. Black contour is the 1% significance level, using a red-noise (autoregressive lag1) background spectrum. **(c)** The global wavelet power spectrum (black line). The dashed line is the significance for the global wavelet spectrum, assuming the same significance level and background spectrum as in (b). Reference: Torrence, C. and G. P. Compo, 1998: A Practical Guide to Wavelet Analysis. Bull. Amer. Meteor. Soc., 79, 61-78.





**Acknowledgements**

This study was primarily motivated by Rhodes W. Fairbridge's work around the idea that the solar system regulates the solar and Earth's climate (Mackey, 2007). However, important motivations have also been coming from several important climate researchers such as: John Sanders, George Kukla, Gerard Bond, and Wallace Broecker, from LDEO/UC, and Charles

Keeling, Timothy Whorf, and Walter Munk from SIO/UCSD, who through their works and words motivated the author to search for cosmic recurrent influences and long-term and lagged climatic/oceanic responses. The author thanks Professor Jan Veizer for his encouraging comments on the first version of this work. The author also thanks Professor Judith Curry for her important recommendations on the last version of this work. The author thanks Jochem Marotzke and Johann Jungclaus, for their AMOC data. The author also thanks Mars Rundgren, for his stomatal-based $CO_2$ reconstructed data. The author also

thanks Jana Schroeder, for her editorial contributions, and Oscar Alonso and Ricardo Espinoza, for their graphical contributions to this work. This work was initiated when the author was supported (2008-2012) by a NSF grant (GEO-0452325) through the IAI project CRN-II-2050 and by the Institute UC MEXUS and CONACYT through an international collaborative project between IMTA and SIO/UCSD under the 2008 Climate Change Program.

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



**Table 1. Lags of different variables of the Earth's climate system with respect to solar activity as a response to the detected 9500-yr solar activity patterns by Sánchez-Sesma (2016)**

| Variable or Model | Area of Application | Lag, with respect to SA [yrs] | Ratio Lag/Model |
|---|---|---|---|
| $CO_2$ Antarctic EPICA | Global Atmosphere | 1670 | 0.985 |
| CO3 in South Atlantic | Global Ocean | 1550 | 0.914 |
| Fe deposition in SW Pacific | Global Ocean | 1570 | 0.926 |
| Volcanic Activity | Global | 780* (=1560/2) | 0.455 (=0.91/2) |
| Power Law Model | Global | 1695 | 1.000 |

- **Although the lag of the volcanic activity is the half of that one of the 9500 yr period, it is considered as a part of the**
5 **Earth's climate system response.**





**Figure 1. Solar and terrestrial climate signals during the deglacial and Holocene. (a) Solar activity, TSI, reconstructed by Solanki et al. (2004; S04), Steinhilber et al. (2009; S09), and Steinhilber et al. (2012; S12), after an intercalibration using the S09 record as a base. (b) A volcanic activity index record based on the Antarctic EPICA dome C ice core sulphate densities (Castellano et al., 2005; C05). (c) Carbonate ion content (CO3) record from the SW Atlantic deep sediments for the last 28 Kyr (Yu et al., 2014; Y14). (d) The continental tropical Congo River basin (CRB) mean annual surface air temperature detrended record (Tcrb) for the last 21000 years, based on lipids (Weijers et al., 2007; W07).**







**Figure 2. Empirical estimation of the lag of ~1600 yrs correspondent to the ~9500 yrs solar activity recurrent pattern (indicated with red mark and color lines), based on multiscale empirical models of climatic lagged response. Three power law models of lags, for the average and +/- standard deviation values, which were adjusted to four estimations made by Shackleton (2000) and van Andel (1994), are graphically and numerically displayed. See data in Table A1, Appendix A.**





**Figure 3. Modeling and forecast of volcanic activity. (a) Volcanic activity index record (C05) modelled as a lagged response (lag=780 yrs) of solar activity (TSI) based on Eq. 2; (b) volcanic activity index record (C05) modelled with the same record transformed linearly as a lagged recurrence (lag=9480yrs) of the same record, based on Eq. 3.**





**Figure 4. Modeling and forecast of a carbon cycle index. (a) Carbonate ion content (CO3) record (Y14) modelled as a lagged response (lag=1550 yrs) of solar activity (TSI) based on Eq. 2; (b) Carbonate ion content (CO3) record (Y14) modelled with the same record, transformed linearly, as a lagged recurrences (lag=9500 and 19000 yrs) of the same record, based on Eq. 3.**





**Figure 5. Cross correlation between the continental tropical Congo River basin (CRB) mean annual surface air temperature detrended record shown in Fig. 1d, and the TSI(S09) lagged record. Peaks around 40, 1650 and 3200 yrs indicate the lags and recurrences between these signals. Periodic (sine) functions for 0.8 and 1.6 Kyr periods are also displayed with dotted lines. The sum of these two functions are displayed with a bold line and explain more than 76% of the evaluated Tcrb-TSI cross-correlation.**







**Figure 6.** Tropical climate (CRB) Tcrb reconstructed and detrended signal and its solar-based models: a) Solar lagged model TSI (S09), b) Sum of three solar lagged models TSI (S09) for 40, 780 and 1600 yrs. The explanation of variance of models a) and b) is 5  **32 and 62%, respectively.**







**Figure 7. Tropical climate (CRB) Tcrb reconstructed and detrended signal and its solar-based models (Only solar direct influences, without solar indirect influences of volcanic and $CO_2$ effects). The Tcrb models, extrapolated backward in time, were based on: a) TSI (S04), b) TSI (S09) and c) TSI (S12), applying Eq. 2 with greater modulated amplitudes before 10Kyr BP.**





**Figure 8. Continental tropical climate reconstructed record, Tcrb, and its analogue model with a lag of 9.6 Kyrs but including two forecasts of TSI, provided by 1) an analogue model TSI (S04) proposed by Sánchez-Sesma (2016; SS16), and 2) the TSI forecast by S13. (a) The CRB T signal and its model, during the last 1 and future 1 Kyr, and (b) a zoom of a) during the last 0.2 and future 0.6 Kyr.**





**Figure 9. CO₂ stomatal-based reconstructed records (RB99, RB03) compared with their analogue model based on the same lagged and adjusted records. a) From 12000 BC to 5000 AD, and b) a zoom of a) from 2000 BC to 3000 yrs AD.**







**Figure 10. CO$_2$ stomatal-based reconstructed record (RB99) compared with its model based on the lagged responses to solar activity (TSI[S09]) and observational record (NOAA, 2016). a) from 0 to 2200 AD, and b) a zoom of a) from 1000 to 2200 AD.**