# Peer review of "The Earth's climate system recurrent & multi-scale lagged responses: empirical law, evidence, consequent solar explanation of recent CO2 increases & preliminary analysis"

_Earth System Dynamics, 2016_

## Author Comment (AC1) · 10 Oct 2016

In Appendix B: [Global ocean circulation (GOC) carries a signal of tropical climate that generates lagged responses of the Earth's climate system] (Attached as Supplement), we analyze the proposed mechanism for the lagged responses of the Earth's climate system to the ∼9500 yr recurrence of solar activity, considering: a) the space-time moving climate variable, Tcrb ("the signal"), b) its lagged influences in the north Atlantic climate and sea level, and c) the global ocean circulation processes ("the carrier") in surface and deep ocean water.

[Figure]

In Appendix C: [Sea level (SL) lags (∼1600 yr) continental tropical climate (CTC)] (Attached as Supplement), we: a) present SL reconstructed records and their verification and adjustment, b) propose and apply a model for SL variations in term of the lagged influences of CTC variations and their verification with their recurrences, and c) present their consequent forecast.

Please also note the supplement to this comment:
http://www.earth-syst-dynam-discuss.net/esd-2016-38/esd-2016-38-AC1-supplement.zip

---

## Author Comment (AC2) · 11 Oct 2016

In this appendix we present evidences of lagged influences of the high northern latitude (HNL) climate [Arctic and northern forests temperature (TA, TNF)] over the Northern Hemisphere Temperature (NHT or T). Specifically, we present: a) reconstructed, simulated and observed records, b) evidences of T lags TA temperatures by multidecadal scales, b) evidences of T lags TNF by multidecadal scales, and c) an intercomparison of their consequent forecasts and a comparison with adjusted and extrapolated climate scenarios considering "zero emissions of CO2."

[Figure]

Please also note the supplement to this comment:
http://www.earth-syst-dynam-discuss.net/esd-2016-38/esd-2016-38-AC2-
supplement.zip

––––––––––––––––––––––––––––––––

---

## Referee Comment (RC1) · Anonymous Referee #1 · 19 Oct 2016

This paper presents a statistical analysis of several environmental and solar proxies. The objective is to outline time lags between terrestrial environmental proxies and solar variability. The paper suffers from a large number of flaws, including in the presentation that prevent me from recommending its publication.

1. Major comments

The presentation is not adequate for a scientific paper. The author cites in extenso various excerpts from publications, as if they were biblical quotations. Most of the citations seem picked out of their original context and are probably not relevant to the

precise analyses carried in the paper. The overall writing yields too much confirmation bias to be adequate for a scientific paper: the author expresses (in a rather confusing way) many general hypotheses, the tests that are made are not very convincing (I will treat this after), and he cites the conjectures formulated by others (and in other contexts) to "confirm" his own hypotheses. Handwaving can be sometimes useful if alternative explanations are formulated. Otherwise, it is rather misleading at best.

The introduction mixes time scales (from millions of years to centennial variability) and tries to convey the impression that of something is valid on a multi-millennial time scale, then it has a form of validity on longer and shorter time scales. The analyses are presented in one-paragraph subsections that just state that a figure shows the results or supports a discussion. The discussion brings other results and further analyses, which are barely enlightened by the appendix (the so-called power laws).

The author claims that his temperature forecast for the 21st century is lower than those of (I guess) CMIP5 simulations. None of the figures/results support such a claim. The results could never support it because the regression is based on millennial averages, and the ongoing climate change is much shorter time scales.

I think that good scientific papers should only marginally deviate from the following structure: (i) statement of the scientific problem with quantitative elements, (ii) presentation of hypotheses, data and methods, (iii) description of results, (iv) discussion, (v) conclusions. Arguments that are not supported by the results of the paper are red herrings. The present manuscript is hence far from the ideal that is recommended by most journals (including ESD).

The data are not presented in a proper way and are used outside of their context. The carbonate ion record is not a proxy for atmospheric $CO_2$. I wonder if the author read the paper of Yu et al. (2014), who focused their discussion on glacial to interglacial changes, which is hardly the subject of the manuscript. There are less than 10 data points between 11 kyr and 0 kyr BP (the common period of all proxy records of the

paper). The detrending of the Congo River data does things that would make most paleoclimatologists wince, because what Weijer et al. (2007) discuss is precisely the trend, the residual being noise from the reconstruction method. The author first states that the Congo temperature is independent of the ocean circulation (p. 6, l. 5), then makes the hypothesis that they are dependent (p. 6, l. 25). Please be consistent. The Congo river data comes from a marine sediment core (Weijer et al. 2007), and the authors of that paper are very clear on the uncertainties of the continental temperature reconstruction. Why not use other high-resolution continuous records of the Holocene?

The simple fact that all the proxies have different time resolutions and variable time samplings precludes the type of analysis that the author wants to make. The author states that he avoids any analysis of uncertainty (e.g. due to dating uncertainties). But he cannot ignore that there are large dating uncertainties in continental proxies and ice core records. Such uncertainties might dominate in the estimation of the parameters of the three models. A "preliminary" analysis is no excuse for not investigating uncertainties, given the strong conclusions that he draws from the analyses. And I cannot encourage the editors of a journal whose impact factor exceeds 4 to publish a paper that does not carry analyses of uncertainty, when the whole topic is about statistical models.

The author has the strange habit of not defining acronyms the first time they are used (e.g. GOC, AMOC, SA?). This is particularly frustrating in the three model equations. What is SC? I understand that P is a proxy variable. Is it one of the datasets shown in Figure 1? What is the magnitude of the error? The formulation of the three models indicates a monotonic trend (the beta term). None of the figures yield such a trend. The value of the parameters is never discussed. The third model is some sort of lagged auto-regressive process. If alpha>=1, beta!=0, gamma!=0, then this model always diverges. This is not very useful.

The results section is based on a further methodological section, with an appendix that brings essential information. Appendixes are there to enlighten some technical issues,

not to carry essential results that are discussed throughout the paper.

What is called a power law has nothing to do with power laws in probabilities or in physics. Eq. (A1) does not make sense. The right hand side no longer depends on t, which is the function argument. What is the relation between this obscure appendix and the three models, which are the gist of the paper?

There are many other problems that could also suffice to reject the paper. Let me conclude this section by a last one. The author makes a "forecast" of atmospheric $CO_2$ for the next century, based on information of the last millennia. It does not seem to have occurred to the author that there are several ways of tracing the origin of the atmospheric $CO_2$. One of the ways is to determine the ratio of stable isotopes in the $CO_2$. It has been known for many years that such measurements point to a fossil fuel origin, not natural variability (or forced by natural causes, like solar activity). There is no chance that atmospheric $CO_2$ decreases that fast after 2100. This simple observation (that can be obtained by reading papers or reviews, such as the various IPCC reports) simply invalidates the model presented by the author to draw conclusions about the future. Basic physics dismisses the whole statistical analysis.

2. Less crucial comments

The bibliographic search of the author seems rather incomplete. He claims that no one looked at the effect of interstellar dust on climate. There were quite a few papers in the 1990's on the subject. They are now barely cited because they did not survive the shock of new observations. The effect of cosmic rays on the stratosphere has been discussed for decades (this is the origin of the cosmogenic isotopes that can be measured at the surface of the Earth).

The author should do a more careful search on the effects of volcanoes on climate, which depend on the time scale (from months to millennia). Since he discusses time scales overs millions of year, he should be aware that volcanoes also expel greenhouse gases like $CO_2$, which eventually warm the troposphere.

The suggestion that a ~9500 year cycle is a harmonic of an excentricity cycle and that it has any link with decadal signals is rather ludicrous. There are no systems (ideal or real, linear or nonlinear) that have such high order measurable harmonics. This whole discussion sounds like numerology or a confirmation bias (I recommend that the author reads the book of D. Kahneman, "Thinking, Fast and Slow", Farrar, Straus and Giroux, New York, 2011).

A true confirmation of the model for prediction would use observed temperature data for the recent period. The author would realize that his model does not predict anything very useful for the 21st century.

The latest IPCC report (in 2013) no longer uses A1 or A2 scenarios, but Representative Concentration Pathway (RCP) scenarios.

The time axes of the figures should have the same format, in order to facilitate the reading and appreciation of results.

The author makes strong statements about what the scientific community should do (i.e. study the climate as an "open system", etc.). I would have appreciated that he makes a more careful bibliographic search. Most of what he emphatically advocates has already been published.

I will stop my review here. I think that each paragraph of the manuscript is debatable, but I afraid that I do not have enough time to dwell on the details.
* * *

---

## Referee Comment (RC2) · Anonymous Referee #2 · 28 Oct 2016

In this study, the author investigates the lagged response of the Earth's climate to various forcings, including solar and volcanic ones, with a particular focus on a presumed 9500-year recurrence pattern. This study is a continuation of a previous study by the same author, in the same journal. Most of the analysis is based on visual comparisons and correlations between various observables.

The quality of this study if well below the standards of rigorous scientific publications: it suffers from a large number of flaws, both technical and methodological. Most claims are lacking statistical evidence, and are not supported by physical explanations. For

that reason I recommend to reject it without revision. Some general comments are listed below, with a few examples.

** Technical comments **

The overall structure of the article is very confusing and the common thread is hard to follow: some elements that are important for understanding the reasoning either are not discussed, or are deferred to the appendix. More generally, too many claims are made without any critical discussion (see methodological comments) and there is a frequent confusion between facts and assumptions.

This problem is aggravated by a poor presentation and sentences that are hard to understand, often because of improper use of technical terms. Here are some examples: - "period" occasionally refers to periodicity and then, to time scales; - "trend" refers to what seems to be a pattern; - "advected" (page 5, line 5) cannot be used to describe solar influence on climate; - "estimates of solar forcing spanning almost an order of magnitude" (page 5, line 3): this is not only vague but also incorrect (different scenarios of the total solar irradiance differ by approximately 1% or less).

On several occasions I have been struggling with English problems, and inconsistencies of the text, trying to understand what was meant. An example of a paragraph that is hard to understand is on page 7, lines 11-15.

There is an indiscriminate use of acronyms (also in figures) some of which are not explained, such as AMOC and GOC.

Several references are not appropriate or are cited out of context, e.g. Haigh et al. (2011) on page 5, line 10. Another one is Peterson (1993) on page 10, lines 8-18. Conversely, some important references are not cited, see below. There are other types of inconsistencies: what is the "climate-ocean group of the Max Planck Institute" (page 8, line 9)?

Most plots are of poor quality, with colour codes that make them hard to read, a mix

of fonts of all sizes, legends that look more like variable names from a computer code, axis units that are unclear, etc. For example, in Fig 3a, what are the units of the volcanic signal ? And in Fig. A1, what do the different equations correspond to ? Note that the coefficients have an unrealistic large number of digits. Equally annoying is the lack of effort to make the plots more easily comparable. Why, for example, not simply use a common time abscissa in Fig. 1 ? In addition, why is Fig. 2 shown twice (same as Fig. A1)?

The title is looooong and not catchy.

** Methodological comments **

Cherry-picking : on several occasions references are selectively chosen among those that support the conclusions of the article, ignoring the numerous other studies that contradict these conclusions, or provide a different view. An example is provided below.

Lack of statistical testing : much of the evidence for lags is based on handwaiving arguments, with no rigorous statistical results to support these. The apparent correlation that is seen in Figure 3 between solar and volcanic signals, for example, is not convincing. The literature is replete of plots showing similar time series, which are then abusively interpreted as causal connections. There are simple tools for estimating such lags between time series, and, more importantly, for determining whether these lags are significant or fortuitous. None of these tests are ever used in this study, which is simply not acceptable, given the conclusions that are drawn from these lags. One example is the correlation "seen" in Figure 3 between solar and volcanic signals.

Putting up a smokescreen where simple critical reasoning is required. In several parts the reasoning is difficult to follow because 1) not all pieces of information are provided (i.e. the results are not reproducible) and/or 2) the reader is misled by information that is uselessly complicated, or simply irrelevant. One example is the power law model (see below) which is not needed to link lags to periods. Another example is the list of models given in Equations 1 to 3, which are unnecessarily complicated for the purpose

of this article.

Inconsistent results and lack of physical foundation : several physical explanations are invoked without clear justification, or without support from other studies. For example, how could one "generate a kind of resonances between lagged responses and oceanic recurrent oscillations" (page 8, line 12) ? And how could a 10:1 resonance with orbital variations give rise to a ~9500 year period ? Note also that the planetary influences that are invoked here have been severely criticized, see for example [Poluianov and Usoskin, Solar Physics, 289 (2014) 2333]. None of these references is cited, thus giving a biased view of the problem.

Errors : on several occasions, the results are simply incorrect. In Figure 4a, for example, the TSI records is replicated several times, as if it were periodic. What allows you to do that ? Another example is the Congo River basin surface air temperature: on page 6, line 5 you state that is independent of ocean circulation. On line 25 you say the opposite. Which is correct ? Note in addition that for detecting weak climate changes, proxies from the intertropical region are preferable to those from the tropical region. The main problem, however, is with the power law model, see below.

Since there is a strong focus on the "power law" model and on the 9500-year periodicity, let me address some issues here.

** Power law model **

First, there is no evidence for a power law scaling between period and lag because your exponent beta is so close to one. You could just as well fit your results with a simple linear model - use Occam's razor.

Second, the model is based on the premise that the physical processes behind these (widely different) time scales should lead to the same response, or at least the same phase lag. This assumption is not substantiated by physical evidence. Actually it makes no sense to compare geological and climatological time scales.
Third, and worse, this model does not tell us anything new. Most quasi-linear systems that are forced by a periodic input will respond with a lag whose value (expressed in terms of the phase) will be between 0 and 2 pi because values larger than 2 pi are ambiguous. Actually, most values should fall between 0 and pi, otherwise one would speak in terms of anticorrelation. So it is quite normal to find most lags between 0 and pi. According to Table A1 your lags range from 0.9 to 1.9 rad. Therefore, there is nothing special with lags that line up linearly with the characteristic time scale. In this sense Figure A1 merely expresses a statistical result with no deep physical meaning. And it certainly does not allow us to estimate the lag associated with the presumed 9500-year oscillation.

Finally, the regression procedure is not detailed. More importantly, the text does not say how the confidence intervals - which are crucial in this context - are derived. One approach would be to use bootstrapping with total least squares. Apparently another procedure was used because I obtain different different confidence intervals. Table A1 mentions empirical error bars for the regressors only; note that the period also has large errors (probably even larger than for the lags), which should be included.

** 9500-year periodicity **

First, how can such a periodicity be meaningfully detected in a solar proxy whose total duration barely exceeds 11000 years ? On such time scales, cosmogenic indices are affected by changes in the geomagnetic field, whose precise evolution is still an active research topic. The long 10Be ice-core records that are used in SS16 cannot be meaningfully interpreted in terms of solar periodicities because changes in atmospheric transport may play an important role here.

Second, several experts in cosmogenic indices have already looked for periodicities in cosmogenic indices, see for example [McCracken et al., Solar Physics 286 (2016) pp. 609-627]. None of them was careless enough to look for periodicities beyond approximately 3000 years. None of these studies finds evidence for a 9500-year period.
Third, in the discussion, when other observables do not show clear evidence for a ~9500-year period, then resonances are invoked to reach that value. Given the large error bars, this means that almost any periodicity can be combined to yield a value that is close to 9500 years. This is not only absurd, but, in addition, no physical justification is given.

---

## Short Comment (SC1) · 30 Nov 2016

This comment deals only with the atmospheric $CO_2$ data plotted in the discussion paper, an issue which - as far as I was able to follow - has not yet been brought up by the reviews published until today.

In Figures 9 and 10 $CO_2$ is plotted based on stomata-based reconstructions for the last 12,000 years (the Holocene) with zoom on the more recent period (last 2000 years) and is then compared with some calculated $CO_2$ values, which seemed to show similar variability.

The chosen $CO_2$ proxy (stomata-based) is known to be a poor recorder of the small scale variability of $CO_2$ we seen during these rather stable time periods. They compare especially weak against ice core $CO_2$ data, which are still believed to be a recorder of ancient atmospheric concentrations. Several papers have discussed these discrepancies and weaknesses of the stomata-based $CO_2$ proxy in detail, (e.g. Ahn et al., 2014; Indermühle et al., 1999; Köhler et al., 2015).

Furthermore, it is well known and established, that the ice core $CO_2$ of the recent past (last 2000 years) from the Law Dome ice core, overlaps without any offset with the instrumental $CO_2$ measurements, which started in year 1958 in Manua Loa, Hawaii, giving firm evidence that ice cores indeed record atmospheric $CO_2$ without any significant offset (MacFarling-Meure et al., 2006; Rubino et al., 2013).

The knowledge on $CO_2$ variability over the last 2000 years has been extended by some $CO_2$ data from the West Antarctic Ice Sheet Divide ice core (Ahn et al., 2012; Bauska et al., 2015). In addition to the Law Dome data we now know that $CO_2$ varied between 270 and 285 ppmv during the last 2000 year, starting to rise due to anthropogenic emissions around year 1750 CE from 278 ppmv to nowadays around 400 ppmv. The variability of $CO_2$ between 270 to 390 based on stomata and plotted in Figure 10 is not supported at all by the more reliable ice core $CO_2$ data.

For the Holocene (last ∼12,000 years) $CO_2$ variability is - again based on ice core data (Monnin et al., 2004; Elsig et al., 2009) - well established to be between 255 and 285 ppmv, consisting of a ∼10 ppmv decrease between 11,000 and 8,000 years before present and a gradual rise thereafter. Non of that is found in the stomata-based $CO_2$ proxy record plotted in Figure 9.

These misfits of plotted $CO_2$ values (based on stomata $CO_2$ proxies) from ice core $CO_2$ are severe shortcomings of the study. If ignored, it would suggest, that our knowledge of $CO_2$ based on ice cores is wrong, for which no futher support is given. For any further details on the difference of ice core based $CO_2$ and stomata-based $CO_2$ please

refer to other papers discussing those in detail (Ahn et al., 2014; Indermühle et al., 1999; Köhler et al., 2015).

**References:**

Ahn, J., Brook, E. J., Mitchell, L., Rosen, J., McConnell, J. R., Taylor, K., Etheridge, D., and Rubino, M.: Atmospheric $CO_2$ over the last 1000 years: A high-resolution record from the West Antarctic Ice Sheet (WAIS) Divide ice core, Global Biogeochemical Cycles, 26, GB2027, doi:10.1029/2011GB004247, 2012.

Ahn, J., Brook, E.J., Buizert, C., Response of atmospheric $CO_2$ to the abrupt cooling event 8200 years ago, Geophys. Res. Lett. 41 (2), 604-609, 2014.

Bauska, T. K., Joos, F., Mix, A. C., Roth, R., Ahn, J., and Brook, E. J.: Links between atmospheric carbon dioxide, the land carbon reservoir and climate over the past millennium, Nature Geoscience, 8, 383–387, doi: 10.1038/ngeo2422, 2015.

Elsig, J.; Schmitt, J.; Leuenberger, D.; Schneider, R.; Eyer, M.; Leuenberger, M.; Joos, F.; Fischer, H., Stocker, T. F. Stable isotope constraints on Holocene carbon cycle changes from an Antarctic ice core, Nature, 461, 507-510, 2009.

Indermühle, A., Stauffer, B., Stocker, T.F., Raynaud, D., Barnola, J.-M., 1999. Early Holocene atmospheric $CO_2$ concentrations, Science, 286 (5446), 1815.

Köhler, P.; Fischer, H.; Schmitt, J.; Brook, E. J., Marcott, S. A. Comment on *"Synchronous records of $pCO_2$ and $\Delta^{14}C$ suggest rapid, ocean-derived $pCO_2$ fluctuations at the onset of the Younger Dryas by Steinthorsdottir et al"*, Quaternary Science Reviews, 107, 267-270, doi: 10.1016/j.quascirev.2014.09.024, 2015.

MacFarling-Meure, C., Etheridge, D., Trudinger, C., Langenfelds, R., van Ommen, T., Smith, A., and Elkins, J.: Law Dome $CO_2$, $CH_4$ and $N_2O$ ice core records extended to 2000 years BP, Geophysical Research Letters, 33, L14810, doi: 10.1029/2006GL026 152, 2006.

[Figure]

Monnin, E., Steig, E. J., Siegenthaler, U., Kawamura, K., Schwander, J., Stauffer, B., Stocker, T. F., Morse, D. L., Barnola, J.-M., Bellier, B., Raynaud, D., and Fischer, H.: Evidence for sustantial accumulation rate variability in Antarctica during the Holocene, through synchronization of $CO_2$ in the Taylor Dome, Dome C and DML ice cores, Earth and Planetary Science Letters, 224, 45–54, 2004.

Rubino, M., Etheridge, D. M., Trudinger, C. M., Allison, C. E., Battle, M. O., Langenfelds, R. L., Steele, L. P., Curran, M., Bender, M., White, J. W. C., Jenk, T. M., Blunier, T., and Francey, R. J.: A revised 1000-year atmospheric $\delta^{13}$C-$CO_2$ record from Law Dome and South Pole, Antarctica, Journal of Geophysical Research: Atmospheres, 118, 8482–8499, doi:10.1002/jgrd.50668, 2013.

———————————————————

---

## Author Comment (AC3) · 30 Nov 2016

Initial Declaration This paper has presented an integration of Earth's climate information over the last millennia and its preliminary analysis, to be discussed, and then improved. In particular, this paper presents evidences of the climate lagged responses both in external and internal forcing (volcanoes, carbon cycle and oceanic transports) and the most important climate variables (Continental Tropical Climate [CTC], Atlantic Meridional Overturning Circulation [AMOC], Sea Level [SL], Ice Raft Debris [IRD] and northern hemisphere and global temperatures [NHT & GT]) in order to better under-

stand and consequently to better model their associated climatic processes and mechanisms.

General Response Thanks to comments and critics, the final version will show, with better structure and information, how the Earth's climate is influenced by the oceanic transport with consequent lagged influences. However, in response to the comments received, in this first reply I will focus on reinforcing the two key points on which my work is based, through new contributions that provide greater solidity, consistency and clarity to the ideas presented in my research.

PART A: Solar recurrences. During recent decades, a great amount of new geophysical and astronomical information has provided us with objective elements to refine our conception of solar dynamics. On the one hand, isotopic information from polar and deep sea sediments has provided a great amount of detailed information for indirect solar activity estimations. On the other hand , solar dynamics simulations, motivated by space explorations, have reconstructed and forecasted millennia scale movements, velocities, and accelerations for the main objects of the Solar System. With these two different but complementary types of information, we have detected low-frequency common variations between solar activity and solar movements. Specifically, in the supplements and based on independent data, we verify the existence of a multi-millennia solar recurrent pattern detected by Sánchez-Sesma (2016a). It is based on data previously published on atmospheric 14C from the Cariaco basin sediments accumulated over the last 60,000 years (Hughen, 2004). Although we have provided [in Appendixes A and B of Sánchez-Sesma (2016a)] further verification, testing and analysis of solar recurrent patterns since geological eras, and their potential gravitational forcing, in this supplementary information, motivated by reviewers 1 and 2, we present additional evidences of: a) the existence of multi-millennial (∼9500-yr) scale solar oscillations, and b) their recurrences. These verified solar recurrent oscillations have also provided additional support not only for the consequent multi-millennial-scale experimental forecast, suggesting a solar decreasing trend toward Grand (Super) Minimum conditions for the upcoming period, 2050-2250 AD (3750-4450 AD), but also for the terrestrial responses analyzed by Sánchez-Sesma (2016b).

PART B: Lagged responses in climate and its Power Law. Even when humans were not impacting the environment, Nature was expressed in dynamic and changing forms. Geology and Biology have shown that Nature's changes usually happen slowly over long time periods that exceed the life span of humans. So, in most cases, we do not see long-term changes in Nature. These unseen changes at one level can influence other levels, cascade down or up levels, reinvigorating or destroying processes that are represented in power-law equations. These power-law equations have been developed as the simplest scale invariant equations in astronomy (Nottale, 1997). In the beginning these equations provided solutions, and a large part of the research on this subject was devoted to the calculation of numerical values of the exponents, but now astronomical researchers know they must go beyond such an approach, and look for the fundamental physical principles that may allow us to really understand how the power-law works (Nottale, 1997). Motivated by these astronomical experiences, and in order to explore deeper in climatic processes, we need as a first step to not only look for these Power-Laws, but also to (accept) analyze these relationships between geophysical scales and processes. We add new information and propose a different approach to analyze the Power-Law variations of the forcing period/response lag (P/L) relationships across different scales. In supplements (one to complement Appendix A), we have applied a second approach to the power law variations of the climatic lagged responses. To do so, we evaluate the Northern hemisphere temperature lag with respect to the annual solar cycle forcing, evaluate a complementary and independent model of the power law based on independent and complemented information, and compare the new approach with the previous approach. The consequent lag associated with the 9.5-kyr solar forcing is 1540 yrs, which is almost 3% lower than the 1586 yrs estimated with the first approach. This reduction, obtained in the second approach, implies a better comparison with other lags associated with the 9.5-kyr, with their mean average ratio changeing from 0.937 in the first approach, to 1.032 in the second approach, implying a reduction to almost half of the differences (6.3 to 3.2 %) to the ideal ratio 1. This supplement will be included in Appendix A.

REFERENCES Nottale L, Scale Relativity, Lecture 19, in Ed. B. Dubrulle, F. Graner and D. Sornette, Scale invariance and beyond, Proceedings of Les Houches school, DAEC, Observatoire de Paris-Meudon, CNRS and Université Paris VII, F-92195 Meudon Cedex, France EDP, Sciences/Springer 1997, pp. 249-261.

Sánchez-Sesma, J.: Evidence of cosmic recurrent and lagged millennia-scale patterns and consequent forecasts: multi-scale responses of solar activity (SA) to planetary gravitational forcing (PGF), Earth Syst. Dynam., 7, 583-595, doi:10.5194/esd-7-25583-2016, 2016a.

Sánchez-Sesma, J.: The Earth's climate system recurrent & multi-scale lagged responses: empirical law, evidence, consequent solar explanation of recent $CO_2$ increases & preliminary analysis, Earth Syst. Dynam. Discuss., doi:10.5194/esd-2016-38, 2016b.

Please also note the supplement to this comment:
http://www.earth-syst-dynam-discuss.net/esd-2016-38/esd-2016-38-AC3-supplement.zip

———————————————